# Unleashing the Full Potential of Product Quantization for Large-Scale Image Retrieval

**Yu Liang**[1,4*]     **Shiliang Zhang**[2]     **Kenli Li**[1†]     **Xiaoyu Wang**[3†]

[1]College of Computer Science and Electronic Engineering, Hunan University
[2]National Key Laboratory for Multimedia Information Processing,
School of Computer Science, Peking University
[3]The Hong Kong University of Science and Technology(Guangzhou)
[4]Intellifusion Inc.
`{leungyu, lkl}@hnu.edu.cn, slzhang.jdl@pku.edu.cn, fanghuaxue@gmail.com`

## Abstract

Due to its promising performance, deep hashing has become a prevalent method for approximate nearest neighbors search (ANNs). However, most of current deep hashing methods are validated on relatively small-scale datasets, leaving potential threats when are applied to large-scale real-world scenarios. Specifically, they can be constrained either by the computational cost due to the large number of training categories and samples, or unsatisfactory accuracy. To tackle those issues, we propose a novel deep hashing framework based on product quantization (PQ). It uses a softmax-based differentiable PQ branch to learn a set of predefined PQ codes of the classes. Our method is easy to implement, does not involve large-scale matrix operations, and learns highly discriminate compact codes. We validate our method on multiple large-scaled datasets, including ImageNet100, ImageNet1K, and Glint360K, where the category size scales from 100 to 360K and sample number scales from 10K to 17 million, respectively. Extensive experiments demonstrate the superiority of our method. Code is available at https://github.com/yuleung/FPPQ.

## 1 Introduction

Approximate Nearest Neighbors Search (ANNs) aims to quickly find necessary information within vast amounts of data. It is commonly used in image and video search [49, 37, 4], recommendation systems [44, 8], and anomaly detection [34, 1] tasks. Hashing [18, 7, 16, 37], which is a key component of ANNs, has been continuously and rapidly developed over time. Hashing enables the efficient mapping of high-dimensional floating-point data into binary code, with the main objective of minimizing storage space and retrieval time while maintaining satisfactory retrieval accuracy.

In the past decade, researchers have been exploring the use of deep learning to optimize hash technology for improved retrieval performance [18, 23, 7, 27, 12, 48]. Compared to traditional methods, deep hashing has shown significant improvement. Nevertheless, to the best of our knowledge, existing deep hash or deep quantization methods have not been comprehensively tested in large-scale data retrieval scenarios, which are commonly encountered in real-world applications. For example, in face recognition [9, 2, 52, 39, 29], where the underlying databases may contain millions of categories and hundreds of millions of samples. The large-scale datasets pose higher requirements to the

---

[*]Work done as an intern at Intellifusion.
[†]Corresponding authors.

37th Conference on Neural Information Processing Systems (NeurIPS 2023).

efficiency of learning and inference, as well as the accuracy of deep hashing algorithms. As shown in our experiments, when tested on large-scale dataset, the classic product quantization (PQ) [20] outperforms many deep hashing methods.

As the one of the most popular methods for quantization retrieval, PQ, however, as shown in Fig. 2, its retrieval performance rapidly declines as the encoding length decreases, rendering it impractical in some applications. Shorter PQ codes can improve retrieval speed and reduce storage overhead, greatly expanding its use, for example, making it a more viable option for edge devices. Due to PQ's widespread use in various industrial applications and its advantages in large-scale data compared to other hash methods, we hope to further unlock the potential of PQ and use it for real-world large-scale image retrieval scenarios. Our objective is to address the challenge that PQ's performance rapidly decays with code length by obtaining a feature representation of the database that is better suited for cluster assignment in multiple sub-spaces. We hold the view that the poor performance of short PQ codes stems from insufficient clustering conditions in the subspace of the original feature vector, resulting in excessively large quantization errors and high error rates during retrieval.

Building on the aforementioned discussion, we propose an end-to-end deep product quantization method called **FPPQ**(**F**ull **P**otential **P**roduct **Q**uantization). Our approach is simple and intuitive: for the large-scale dataset, we generate a PQ code for each class, which is then utilized as a label for learning via a differentiable multi-segment fully connected branch. Upon completing training, the retrieval process is precisely the same as that of traditional PQ, allowing for easy integration into current mature retrieval modules.

The contribution of this paper can be summarized as three fold. **Firstly**, for the first time, we investigate the retrieval performance of current popular or advanced deep hash methods under extremely large datasets with an extensive number of categories and demonstrate that they may not be suitable for such scenarios. **Secondly**, we propose a simple yet effective end-to-end deep feature quantization compression method, based on class-level product quantization coding supervision. Our method achieves a low-slope decay of retrieval performance with decreasing code lengths and can be used for retrieval in the same way as the traditional PQ. **Finally**, we conducted extensive experiments on Glint360K [2], a large-scale datasets with 360k classes and 17 million images, as well as two relatively smaller scale datasets, Imagenet1K [35] and Imagenet100 [7, 35]. The results demonstrated the effectiveness of our method.

## 2 Related Works

Hash technology is widely used for compressing data and speeding up retrieval processes in computer vision. The goal is typically to ensure that samples that are similar in their original space remain similar when they are mapped to the new hash space. Researchers have conducted extensive research on hash technology in recent years, and hash methods can be classified in several ways.

For example, hash methods can be categorized according to the calculation method of hash distance. Hamming distance-based methods aim to learn a binary code and calculate the distance between samples using the Hamming distance metric [16, 19, 32, 42, 46, 18, 37, 12, 25, 24, 40, 7, 36, 26]. Conversely, dictionary-based methods calculate distance by looking up a table through a codebook [20, 51, 15, 28, 13, 27, 23, 49, 45].

In an alternative way, hash methods can be categorized into non-deep and deep methods. Non-deep methods such as LSH [16], Composite Quantization (CQ) [50], Additive Quantization (AQ) [3], PQ [20], OPQ [14], LOPQ [21] are widely used for efficient retrieval. LSH is a data-independent random locality sensitive hash algorithm that can use a variety of methods such as random projection or buckets to directly perform hashing. PQ clusters the sub-vectors and represents original vectors by the cluster categories. OPQ and LOPQ improve upon PQ by transforming the original data to reduce quantization error and considering the local relationships of the vectors, respectively. Non-deep hashing methods have higher efficiency but lower performance, while deep hash methods leverage the powerful representation learning abilities of neural networks to learn hash codes and feature representations simultaneously, reducing quantization error for better performance [18, 49, 7, 37, 23, 45, 27, 47, 48, 51]. Since the quantization process is not differentiable, researchers often employ the *sigmoid* function or *tanh* function to fit the *sign* function [7], or use the straight-through (ST) estimator [5] to avoid gradient non-transfer, as in the case of DPQ [23] and GreedyHash [37]. For more detailed information on deep hashing, please refer to [30].

**Limitation:** Despite the existence of many hashing methods with their own unique advantages, they unfortunately have limitations when applied to large-scale datasets. For example, in the face of large-scale data, the Gini Impurity related penalty constraint item designed by DPQ [23] is difficult to apply and can easily cause network collapse, leading to a large number of data being mapped to the same code. In terms of computing resources, PQN [45] and DTQ [27] are computationally expensive due to the construction of triples, despite DTQ proposal to reduce the amount of calculation by dividing the dataset into multiple groups. DQN [47] requires frequent clustering of the entire training data to update the hash code center. Recent works attempt to apply deep hashing to large-scale datasets with thousands of classes, such as OPQN [49], ADSVQ [51], DCDH [48], etc. However, ADSVQ and DCDH need to calculate a square matrix of the number of categories in the training process, which limits the further expansion of the number of categories in the dataset. The number of bits in OPQN is limited to the feature dimension, resulting in significant training overhead when a longer bit number is needed to ensure retrieval performance. In contrast, our method does not involve large-scale matrix operations, converges quickly, and overcomes the limitations of previous methods for large-scale datasets.

## 3 Preliminaries

Denoting the input images set as $\mathcal{I}$ and the deep features set extracted by the backbone as $F = \{\mathcal{F} \mid \mathcal{F} \in \mathbb{R}^D\}$, Product Quantization divides the feature $\mathcal{F}$ into $M$ segments, i.e., $\mathcal{F} = [\mathcal{F}_1, \cdots, \mathcal{F}_m, \cdots, \mathcal{F}_M]$, where $\mathcal{F}_m \in \mathbb{R}^{D/M}$. Within each segment, all sub-vectors $\{\mathcal{F}_m\}$ are clustered into $K$ categories using the k-means algorithm. Consequently, each feature can be represented as an $M$-tuple: $PQ(\mathcal{F}) = [k_1, \cdots, k_m, \cdots, k_M]$, where each $k_m$ represents the clustering category(index) of the $m$-th sub-vector and ranges from 0 to $2^b - 1$. The codebook $\mathcal{C} \in \mathbb{R}^{M \times K \times (D/M)}$ consist of the clustering centers in each of the $M$ segments, which are used to reconstruct the original features from the $M$-tuple representation $PQ(\mathcal{F})$:

$$quant(\mathcal{F}) = \left[ c_{k_1}^1, \cdots, c_{k_m}^m, \cdots, c_{k_M}^M \right], \tag{1}$$

where $quant(\mathcal{F})$ is the quantized vector that approximates the original feature $\mathcal{F}$, and $c_{k_m}^m$ denotes the $k_m$-th codeword in the $m$-th part of the codebook. Finally, the feature is compressed into $M \times b$ bits.

In retrieval phase, a Lookup Tables (LUTs) will be calculated, which stroed the distances between the query and the codebook elements. Then, the distances between the query and all samples in the gallery set can be efficiently computed by looking up the LUTs.

## 4 Method

### 4.1 Overview

The original PQ method is designed to minimize quantization error using clustering algorithms. However, when we increase the compression strength to reduce the encoding bits in PQ, the number of segments M decreases, and the sub-vector dimension increases, which brings problems: 1) The impact of misallocation of segment increases: the quantization error will increases with the sub-vector dimension increases, and also the importance of a single segment will increases with the number of segments decreases. 2) As the number of segments decreases and the sub-vector dimension increases, the sub-vectors become more like complete features, usually with stronger discriminative. Therefore, the sub-vector distribution becomes more uniform. Since we need to assign $N$ classes to $K$ clustering centers, where $N \gg K = 2^b$, this makes it more likely for features of different classes to be assigned to the same PQ codes, or for the same class to be assigned to different PQ codes. These problems will significantly affect PQ performance.

To mitigate those issues, we train a backbone to generate features that are better suited for a short PQ encoding, Our aim is to assign features of the same class to the same PQ code and cluster the sub-vectors that belong to the same encoding within each sub-space. To achieve these goals, we undertook these two actions: 1) learning a backbone to generate features that better suited for short PQ encoding based on a set of pre-defined class-level PQ code supervision, and 2) using the learned adaptable codebook to guide the PQ encoding process.

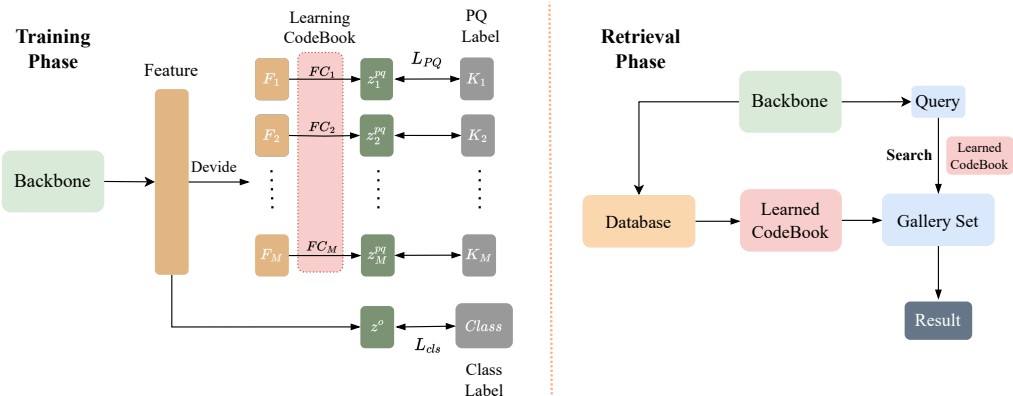

Figure 1: The overall flow of our framework. It consists two branches during the training phase: a one-hot classification branch for the entire feature and another PQ classification branch that constrains sub-vectors of the feature. We use the predefined class-level PQ labels to guide the learning of the PQ branch and improve feature separation in sub-spaces. During the retrieval phase, the only difference with traditional PQ is that our framework does not require clustering for the codebook. Instead, we directly use the learned codebook obtained from the PQ branch, which comprises the fully connected weights of multiple sub-branches in the PQ branch.

Fig. 1 illustrates the overall flow of our framework. We incorporate a PQ branch at the end of the backbone and run it in parallel with the classifier. To learn the PQ code targets for each class, we divide the features $\mathcal{F}$ into $M$ segments refer to the PQ encoding process. For each PQ sub-branch, we maximize the posterior probability of the ground-truth:

$$L_{pq} = -\frac{1}{BM} \sum_{b=1}^{B} \sum_{m=1}^{M} log \frac{e^{z_{mk^l}}}{\sum_{k=1}^{K} e^{z_{mk}}}, \tag{2}$$

where $B$ is the batch size, $M$ is the number of PQ code segments, and $K$ is the number of clusters in each segment. $z_{mk}$ is the output of the $k$-th unit in the $m$-th PQ sub-branch. $k^l$ is the ground-truth value, which is the code value of a unit in the PQ label.

**Optimizing Sub-branches with Angle:** Let us consider the Euclidean distance between the sub-vector $\mathcal{F}_m$ in the $m$-th segment and $K$ classification weights $\{w_{mk}; k \in [1, K]\}$ belonging to the $m$-th PQ sub-branch:

$$D_{Euclidean} \langle \mathcal{F}_m, w_{mk} \rangle = \sqrt{\|\mathcal{F}_m\|^2 + \|w_{mk}\|^2 - 2\|\mathcal{F}_m\|\|w_{mk}\|\cos\theta_{mk}}, \tag{3}$$

where $\theta_{mk}$ is the angle between $w_{mk}$ and the sub-vector $\mathcal{F}_m$. We normalize $\{w_{mk}\}$ by setting $\|w_{mk}\| = 1$, Then, Eq. (3) can be interpreted as:

$$D_{Euclidean} \langle \mathcal{F}_m, w_{mk} \rangle = \sqrt{\|\mathcal{F}_m\|^2 + 1 - 2\|\mathcal{F}_m\|\cos\theta_{mk}}. \tag{4}$$

Now, the ranking of the Euclidean distance between the sub-vector $\mathcal{F}_m$ in the $m$-th segment and all the weights $\{w_{mk}\}$ in the corresponding PQ sub-branch depends solely on the angles $\{\theta_{mk}; \theta \in [0, \pi]\}$. This is because the norm of $\mathcal{F}_m$, denoted $\|\mathcal{F}_m\|$, is invariant to changes in the magnitude of the weights $\{w_{mk}\}$. Exploiting this property, we can optimize M softmax sub-branches based on the angles. To further stabilize the training process and accelerate model convergence, we omit the bias term in the FC layer. Additionally, we normalize $\mathcal{F}_m$ and set $\|\mathcal{F}_m\| = 1$. This get:

$$z_{mk} = \|\mathcal{F}_m\|\|w_{mk}\|\cos\theta_{mk} = \cos\theta_{mk}. \tag{5}$$

Retrieval tasks typically demand a higher degree of discriminability than classification tasks. A simple softmax cross-entropy loss may fall short of achieving the desired results. To enhance the discriminability of each sub-branch, a margin can be added to the loss. Fortunately, we can utilize existing metric learning losses [39, 9, 29] such as CosFace [39] to successfully achieve our objectives:

$$L_{pq} = -\frac{1}{BM} \sum_{b=1}^{B} \sum_{m=1}^{M} log \frac{e^{s(cos(\theta_{mk^l} + margin))}}{e^{s(cos(\theta_{mk^l} + margin))} + \sum_{k=1, k \neq k^l}^{K} e^{s\, cos(\theta_{mk})}}, \tag{6}$$

where $margin \geq 0$ is a fixed parameter to control the magnitude of the cosine margin, and $s$ is a scaling factor.

Apart from the PQ branch, the classification branch is also utilized to enhance the discriminability of the full features. The final optimization objective is defined as:

$$L = L_{cls} + L_{pq}, \tag{7}$$

where $L_{cls}$ denotes any classification loss used to learn the backbone, and we can achieve the optimization objective in an end-to-end manner.

## 4.2 Predefined Class-Level PQ Label

Earlier works have utilized predefined hash codes as labels for guiding model learning, such as DPN [12], CSQ [46], and OrthoHash [18]. They constructed hash codes via Hadamard matrix [41] or Bernoulli sampling to uniformly distribute the hash codes in the target space and maximize the distance between the hash codes. However, as the number of data categories increases, the large dimensions of Hadamard matrix make it impractical to use, and achieving near-uniformity of compressed hash codes based on Bernoulli sampling as a supervised target can be difficult to converge.

In our approach, we get the class average featurew $\{\mathcal{F}_{avg}\}$ for each class by averaging the output features of a pre-trained model. Here, we assume that the classifier of the pre-trained model has been discarded and is not usable, as this is a common scenario, particularly in retrieval tasks. We then execute the PQ encoding process on $\{\mathcal{F}_{avg}\}$ by performing k-means [31] clustering in each sub-space to generate the corresponding class-level PQ encoding. These PQ codes implicitly capture inter-class relationships:

$$
\begin{aligned}
PQ_{label} &= PQ(\{\mathcal{F}_{avg}\}) \\
&= \{[k_1, \cdots, k_m, \cdots, k_M]_{cls} \, ; \, cls \in [1, class\_num]\} .
\end{aligned} \tag{8}
$$

In situations with a lot of data categories and low encoding bits, there is a risk of PQ code duplication. To remove duplication and ensure meaningful inter-class relationships, we remove such duplicates by assigning them to the position with the lowest quantization error, provided that the position is not already occupied by another PQ code.

## 4.3 Encode and Retrieval

After completing the training phase of our framework, we utilize the backbone to extract the feature representation of the input images $\mathcal{I}$. The PQ branch weights, which consist of $M$ FC layers, are used as the codebook in the PQ algorithm to obtain the PQ code of the gallery set and calculate the quantized distance. Specifically, we extract the weights of the PQ branch and perform normalization on them as follows: $\widetilde{w}_{mk} \leftarrow w_{mk} / \|w_{mk}\| ; \widetilde{w}_{mk} \in \mathbb{R}^{D/M}$. All $\widetilde{w}_{mk}$ of the $M \times K$ items in the $M$ branches are combined to a codebook, denoted as $\mathcal{C} \in \mathbb{R}^{M \times K \times (D/M)}$, where $\mathcal{C}_{mk} = \widetilde{w}_{mk}$. At this point, the PQ algorithm no longer requires training, and the codebook $\mathcal{C}$ learned during the training phase will be used to initialize the PQ algorithm. We can encode the samples of the gallery set as $PQ(\mathcal{F}_{\mathcal{G}}) = \{[k_1, \cdots, k_m, \cdots, k_M]_i \, ; \, i \in [1, num\_images]\}$.

There are two methods of utilizing PQ for retrieval: symmetric retrieval and asymmetric retrieval. When apply symmetric retrieval, the distance between samples in the query set and the gallery set is calculated as follows:

$$\mathcal{D}_{sym} = \sum_{m=1}^{M} \langle \mathcal{C}_{mk^*}, \mathcal{C}_{mk^i} \rangle, \tag{9}$$

where $k^*$ is the index of the closest codeword to sub-vector $\mathcal{F}_m$ among the $K$ codewords in the $m$-th part of the codebook, and $\mathcal{C}_{mk^*}$ and $\mathcal{C}_{mk^i}$ represent the quantized representation of the $m$-th segment of the query sample and the gallery sample , respectively. When apply asymmetric retrieval, the distance between samples in the query set and gallery set is calculated as follows:

$$\mathcal{D}_{asym} = \sum_{m=1}^{M} \langle \mathcal{F}_m^{query}, \mathcal{C}_{mk^i} \rangle, \tag{10}$$

where $\mathcal{F}_m^{query}$ is the sub-vector of the $m$-th segment of the query sample. Considering the discussion about Eq. (4), we have:

$$
\begin{aligned}
k^* &= \underset{k\in[0,K-1]}{\arg\min} \left\langle \mathcal{F}_m,\, \mathcal{C}_{mk} \right\rangle \\
&= \underset{k\in[0,K-1]}{\arg\min} \left\langle \widetilde{\mathcal{F}}_m,\, \mathcal{C}_{mk} \right\rangle,
\end{aligned}
\tag{11}
$$

where $\widetilde{\mathcal{F}}_m$ is the normalized vector of $\mathcal{F}_m$ and $\langle\bullet\rangle$ can be either the cosine distance or the Euclidean distance. Referring to Eq. (11), for symmetric retrieval, no additional feature operations are necessary for encoding, even though we perform segment normalization on the features during training phase. When apply asymmetric retrieval, $\sum_{m=1}^{M} \left\langle \mathcal{F}_m^{query}, \mathcal{C}_{mk^i} \right\rangle \neq \sum_{m=1}^{M} \left\langle \widetilde{\mathcal{F}}_m^{query}, \mathcal{C}_{mk^i} \right\rangle$, the retrieved results may be different. However, as $\mathcal{C}_{mk^i}$ is a certain value and the classification loss with margin of PQ branch increases the compactness of the encoding, we found in our experiments that whether applying segment normalization operations for the feature had little impact on the ultimate retrieval performance. The same conclusion holds for the normalization of full features, despite the intuition that normalizing the complete features can enhance the stability of asymmetric retrieval results by fixing the magnitude of $\|\mathcal{F}_m\|$. Overall, whether applying symmetric or asymmetric retrieval, we do not need to perform any additional processing on the features compared with PQ algorithm.

# 5 Experiment

## 5.1 Datasets and Evaluation

Although our primary focus is to evaluate the performance of the proposed hash method on large-scale datasets, we will validate our approach on datasets of various scales, including a large-scale datasets Glint360K, as well as two relatively smaller scale datasets, ImageNet100 and ImageNet1K.

**Glint360K [2]:** Glint360K is one of the largest publicly available face datasets, with over 360K IDs and 17 million images. We use all available data to train hash methods and test for convergence by evaluating the unseen retrieval performance on two other datasets, MegaFace [22] and FaceScrub [33], which are non-intersecting with Glint360K.

**MageFace and FaceScrub:** MegaFace is a dataset of one million images that capture over 690K different individuals and will be used as the gallery set. FaceScrub comprises over 100K photos of 530 celebrities. Following [22], the evaluation was performed on a subset of FaceScrub that includes 80 individuals (40 females and 40 males) from individuals with more than 50 images each. We followed the recommendations from insightface project[1] to remove noise and duplicate samples. The final query set consists of 3529 face images from 80 different individuals.

**ImageNet1K [35] and ImageNet100:** ImageNet1K is a widely-used large-scale visual recognition challenge dataset, which contains 1000 classes and over 1.2 million images. We employed the training set of ImageNet1K to train the hash methods and used the validation set, which consists of 50,000 images, to evaluate hash retrieval performance. Specifically, we selected the top 5 images from each class in the validation set, totalling 5000 images, to construct the query set, while the remaining 45,000 images comprised the gallery set. ImageNet100 has been popularly used in previous deep hashing methods [46, 18, 7], and it is the subset of ImageNet1K, consisting of 100 classes.

**Evaluation:** In the following experiments, we will use asymmetric retrieval and our method will not perform any feature preprocessing during retrieval. We will assess the retrieval performance using Top-1, Top-5, and Top-20 accuracies on the Glint360K datasets. Specifically, the query set consists of N individuals, and for each individual, we have M images. We will test each image by incorporating it into the gallery set and employing each of the other M-1 images as the query set. Regarding ImageNet1K and ImageNet100, we will assess the retrieval performance by using the mean average precision(mAP) of the top 1000 result: mAP@1000.

## 5.2 Implementation Details

All experiments were conducted using the PyTorch framework and 8 NVIDIA 2080Ti or 3090Ti GPUs for training. We employed a pre-trained model to get features so that we could acquire

---

[1]https://github.com/deepinsight/insightface

Table 1: On the large-scale dataset Glint360K, the performance comparison of multiple methods under multiple code lengths, it should be noted that some methods were not listed due to ineffective performance or requiring unrealistic computational resources.

| | Glint360K iResnet50 | | | | | | | | |
|---|---|---|---|---|---|---|---|---|---|
| Method | 32 bits | | | 64 bits | | | 128 bits | | |
| | Top-1 | Top-5 | Top-20 | Top-1 | Top-5 | Top-20 | Top-1 | Top-5 | Top-20 |
| PQ [20] | 0.1724 | 0.2999 | 0.4254 | 0.6665 | 0.7749 | 0.8404 | 0.9184 | 0.9492 | 0.9610 |
| HHF [42] | 0.0001 | 0.0014 | 0.0054 | 0.0773 | 0.1646 | 0.2661 | 0 | 0 | 0 |
| CSQ [46] | 0.0061 | 0.0113 | 0.0183 | 0.0124 | 0.0197 | 0.0269 | 0.0283 | 0.0372 | 0.0451 |
| OrthoHash [18] | 0.3098 | 0.3706 | 0.4171 | 0.5526 | 0.6007 | 0.6351 | 0.6626 | 0.7104 | 0.7454 |
| GreedyHash [37] | 0.3688 | 0.5220 | 0.6323 | 0.6102 | 0.7452 | 0.8251 | 0.8259 | 0.9003 | 0.9338 |
| FPPQ(Ours) | **0.9343** | **0.9601** | **0.9652** | **0.9467** | **0.9571** | **0.9628** | **0.9568** | **0.9690** | **0.9731** |

class-level PQ labels. It is worth noting that this operation is fair since other comparative methods also use this pre-trained model to initialize their model parameters. In all experiments, we set the hyperparameters $s$ to 64 and $margin$ to 0.2. Moreover, we set 8 bits per segment, which is the commonly used setting in practical applications. This implies that the number of categories $K$ per sub-branch was set to 256, while the number of segments varied depending on the target bits. For example, the bits [16-bits, 32-bits, 64-bits, 128-bits] corresponded to the segments [2, 4, 8, 16]. We compared various hashing methods [20, 7, 46, 12, 25, 48, 36, 37, 18, 23, 42, 43], and we implemented these methods based on the cisip-FIRe project[2] and integrated some public projects provided by the specific author. For a fair comparison, we use the default settings in the cisip-FIRe project or used the default hyperparameters specified in their papers, and we trained all methods with the same backbone, same epochs, and same batch sizes, among others. In general, unless otherwise specified, these methods we compare are trained using fine-tuning based on the pre-trained model, and the hash layer had an initial learning rate of 0.0001 that decreased to 0.00001 after training 0.8 times number of epochs. The backbone learning rate was always 0.1 times that of the hash layer. In the following experiments, we will provide more specific explanations regarding implementation details.

## 5.3 Results on Large-Scale Dataset Glint360K

**Implementation Details Supplement:** We employed iResnet [11] as the backbone and adopted the default training settings from the insightface project. The dimension of features output by the backbone was set to 512. We employed SGD as the optimizer with a weight decay of 0.0001. The batch size was set to $128 \times 8$, and the model was trained for a total of 20 epochs. The initial learning rate was set to 0.1 and linearly decreased it to 0 based on the number of training steps.

**Comparison:** We explored and tested various advanced methods, including PQ [20], HHF [42], CSQ [46], OrthoHash [18], GreedyHash [37], JMLH [36], DPQ [23], DCDH [48], DWDM [10], etc. These methods can be considered as encompassing the current most competitive hashing methods. However, DCDH and DWDM were abandoned due to exceeding computational resource limitations with large-scale matrix multiplication, JMLH and DPQ did not achieve effective performance. The further restricted methods are explained in Sec. 2. It should be noted that OrthoHash also applies a similar angular loss function for model optimization. To enhance its feasibility on large-scale data, we set its hyperparameters $s$ to 64 and $margin$ to 0.2, the reason please refer to [39]. For methods that used predefined hash centers [46, 18], we used Bernoulli sampling and ensure that hash centers were non-redundant. This approach is chosen due to the inefficiency and infeasibility of generating hundreds of thousands, or even millions, of hash centers using the technique proposed in OrthoHash. We only demonstrate methods that achieved a certain level of performance in Tab. 1.

As it is apparent, currently available methods fail to achieve satisfactory performance on large-scale datasets under multiple bit settings. The hyperparameters in HHF change depending on the minimum distance bound tables proposed in [6] for different code lengths, which hinders its efficacy to converge in 128 bits. While OrthoHash and GreedyHash have achieved a certain level of performance, they have exhibited inferiority to the original PQ method in both 64 bits and 128 bits. In contrast, our proposed method has displayed significant performance improvement under multiple bit settings.

---

[2]https://github.com/CISiPLab/cisip-FIRe

Table 2: Under different network structure and bits settings, our method achieved consistent and significant improvements.

| Glint360K | | | | | | | | | | |
|---|---|---|---|---|---|---|---|---|---|---|
| Backbone | | iResnet18 | | | iResnet50 | | | iResnet100 | | |
| Method | Metric | 128 bits | 64 bits | 32 bits | 128 bits | 64 bits | 32 bits | 128 bits | 64 bits | 32 bits |
| PQ [20] | Top-1 | 0.8005 | 0.4588 | 0.0973 | 0.9184 | 0.6665 | 0.1724 | 0.9465 | 0.7380 | 0.2074 |
| | Top-5 | 0.8659 | 0.5958 | 0.1940 | 0.9492 | 0.7749 | 0.2999 | 0.9658 | 0.8378 | 0.3569 |
| | Top-20 | 0.9025 | 0.6903 | 0.3031 | 0.9610 | 0.8404 | 0.4254 | 0.9720 | 0.8878 | 0.4941 |
| FPPQ(Ours) | Top-1 | **0.8231** | **0.7019** | **0.6945** | **0.9568** | **0.9467** | **0.9342** | **0.9679** | **0.9578** | **0.9305** |
| | Top-5 | **0.8824** | **0.7890** | **0.7576** | **0.9690** | **0.9571** | **0.9601** | **0.9746** | **0.9728** | **0.9700** |
| | Top-20 | **0.9133** | **0.8414** | **0.8006** | **0.9731** | **0.9628** | **0.9652** | **0.9772** | **0.9753** | **0.9734** |

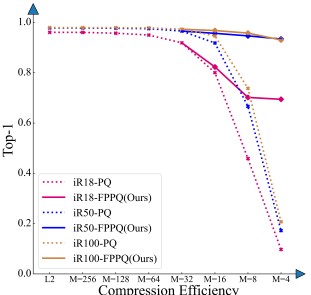

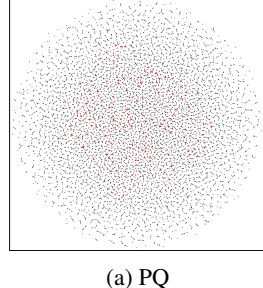

(a) PQ

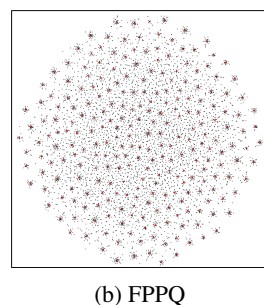

(b) FPPQ

Figure 2: In Glint360K, comparison of the asymmetric retrieval performance trend of PQ with compression efficiency and the results of our methods for improving quality. 'L2' represents using the original features for L2 retrieval, while 'M=256' represents dividing each feature into 256 segments for PQ retrieval, and 8 bits per segment.

Figure 3: The visualization of the sub-vectors and clustering centers for the 10k classes in the Glint360K dataset of two approaches: (a) classification loss with PQ and (b) our proposed method. The black dots represent sub-features, while the red dots in (a) depict the clustering centers obtained by performing k-means clustering on the sub-features of all classes. In contrast, in (b), the red dots represent the codewords trained by our proposed method.

Specifically, in the 32-bit setting, our method outperformed GreedyHash with a nearly 57% increase in Top-1 performance. Additionally, we were able to achieve approximately 28% and nearly 4% performance growth in the 64 bits and 128 bits settings, respectively.

**Performance with Different Backbone Settings:** We also tested the effectiveness of our method under multiple network structures, and our method achieved significant advantages in all results, as shown in Tab. 2. Noting that the large-scale datasets provided favorable conditions for PQ clustering, making PQ a strong baseline, as evident in Tab. 1. Therefore, in this comparison, we focused on comparing our method with PQ (additional results can be found in the supplementary material). We can observe that larger backbones typically result in better performance. However, iResnet100 did not consistent improvement in comparison to iResnet50 when encoded to 32 bits (Top-1↓, Top-5↑, Top-20↑). we suspect that the performance limits of the 32 bits setting may be due to the intrinsic constraint of compression strength.

**Low-Slope Decay of Retrieval Performance:** As the encoding bit-length decreases, the performance of naive product quantization encoding experiences a significant decline. However, after being trained using our proposed method, the performance degradation gradually occurs at a lower slope with decreasing bit-lengths due to our approach's performance improvement at low bit-lengths. This trend is illustrated in Fig. 2.

**t-SNE Visualization:** To better illustrate the effectiveness of our proposed method, we utilize t-SNE [38] to visualize changes in sub-vectors when the feature are compressed to 32 bits. For visualization purposes, We select the first segment of the average features of the first 10,000 classes in Glint360K, denoted as $\{\mathcal{F}_1\} \in \mathbb{R}^{10000 \times (D/M)}$, and their corresponding codewords for the first part, represented by $\mathcal{C}_1 \in \mathbb{R}^{K \times (D/M)}$. As we can see in Fig. 3, the sub-vectors of the original features tend to be

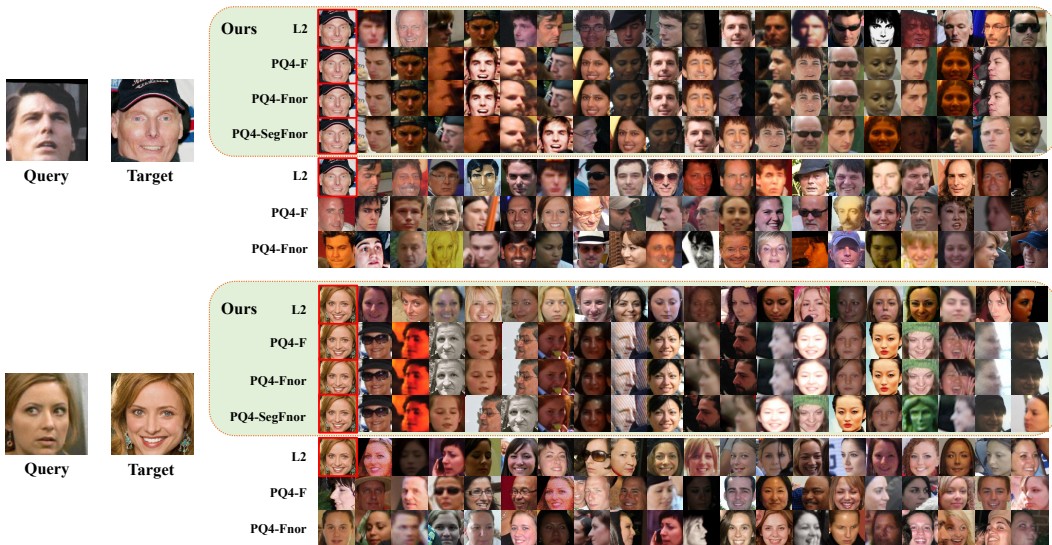

Figure 4: Visualization of the Top-20 retrieval samples using FaceScrub as the query set and MegaFace as the gallery set.

evenly distributed throughout the subspace, hindering sub-vectors compression. In contrast, our method effectively learns class-level PQ labels, leading the sub-vectors to cluster into $K$ clusters in the subspace.

**Retrieval Result Visualization:** Moreover, we provide visualizations of the Top-20 retrieval results on the MegaFace and FaceScrub datasets in Fig. 4. This encompasses comparisons between various ways: 1) our method, which includes L2 retrieval and PQ4 retrieval (without feature processing, normalization, or segment normalization), and 2) the original L2 retrieval and naive PQ4 retrieval (with or without feature normalization). It can be observed that our method generates similar top-20 results under different feature preprocessing settings, further validating the insensitivity of our method to feature preprocessing.

## 5.4 Results on Smaller Scale Datasets

In addition to conducting experiments on large-scale datasets, we evaluated our proposed method on relatively small datasets such as ImageNet1K and ImageNet100. We utilized ResNet50 [17] as the backbone and employed parameters pre-trained on ImageNet1K to initialize the model. The dimension of the features output by the backbone is 2048. Our experiments on these datasets not only demonstrated the effectiveness of our proposed method but also allowed us to make preliminary attempts at addressing two special scenarios.

**Results on ImageNet100:** The experiment on ImageNet100 highlights a scenario where the data scale is small and has fewer categories compared to the number of clusters required for k-means clustering. As k-means clustering cannot be executed in this scenario, we overcame this limitation by training the PQ using all instance features and subsequently encoding the class average features. To prevent overfitting due to the limited data, we fixed the backbone and added a linear layer of 2048 to 128 at the end of the backbone for learning. We fine-tuned our model for 100 epochs with a learning rate of 0.0001. The results presented in Tab. 3, and the performances of other methods were obtained either by replicating the results using the cisip-FIRe project or citing their respective papers. Our method achieved the best performance on ImageNet100, indicating its efficacy in scenarios involving limited dataset scale.

**Results on ImageNet1K:** The experiments on ImageNet1K reveal another scenario: the inability to obtain a good class-level PQ label due to limitations in the number of training samples or low feature discrimination. With only 1000 classes in ImageNet1K, training PQ effectively becomes challenging given the limited number of categories available. Moreover, the low discriminability between classes in the features generated by a pre-trained model leads to a significant amount of duplicate PQ codes

Table 3: mAP@R1000 on ImageNet100

| Method | ImageNet100 mAP@R1000 | | |
|---|---|---|---|
| | 16 bits | 32 bits | 64 bits |
| HashNet [7] | 0.5101 | 0.7059 | 0.7997 |
| CSQ [46] | 0.8379 | 0.8750 | 0.8874 |
| DPN [12] | 0.8543 | 0.8799 | 0.8927 |
| DFH [25] | 0.8352 | 0.8781 | 0.8849 |
| DCDH [48] | 0.7856 | 0.8158 | 0.8400 |
| JMLH [36] | 0.8366 | 0.8671 | 0.8799 |
| GreedyHash [37] | 0.8544 | 0.8796 | 0.8886 |
| HBS-RL [43] | 0.8465 | 0.8702 | 0.8851 |
| DPQ [23] | 0.8860 | 0.8770 | 0.8660 |
| HHF [42] | 0.8710 | 0.8910 | 0.8960 |
| OrthoHash [18] | 0.8693 | 0.8869 | 0.8994 |
| FPPQ(Ours) | **0.8956** | **0.9128** | **0.9154** |

after PQ training and encoding. For instance, duplicate rates can exceed 50% for the PQ codes of 1000 class average features when compressed to 32 bits. To tackle this issue, we adopt the method proposed by OrthoHash, which involves generating hash codes using repeated Bernoulli sampling to achieve binary encodings with sufficient hamming distance. These binary encodings are then segmented and converted into decimal codes to serve as PQ labels. We adopts the default training strategy from the cisip-FIRe project, fine-tuning for 90 epochs. we compared several of the most competitive methods and the experimental results are presented in Tab. 4. As shown in the results, our method achieved competitive results, particularly when the encoding length was relatively short.

Table 4: mAP@R1000 on ImageNet1K

| Method | ImageNet1K mAP@R1000 | | |
|---|---|---|---|
| | 16 bits | 32 bits | 64 bits |
| HHF [42] | 0.2961 | 0.5979 | 0.6121 |
| DCDH [48] | 0.3666 | 0.4764 | 0.5299 |
| CSQ [46] | 0.5040 | 0.6061 | 0.6093 |
| JMLH [36] | 0.5286 | 0.5876 | 0.6098 |
| GreedyHash [37] | 0.5424 | 0.5896 | 0.5952 |
| OrthoHash [18] | 0.5936 | 0.6514 | **0.6761** |
| FPPQ(Ours) | **0.6207** | **0.6543** | 0.6649 |

## 6 Conclusion

This study explored and evaluated several advanced deep hashing methods for large-scale image retrieval, identifying their limitations. In addition, we proposed a product quantization-based framework that employs multiple softmax branches to learn specific class-level PQ labels. Our proposed method effectively performs during the retrieval stage and aligns well with the traditional product quantization methods. Notably, our method is independent of the number of classes and data samples required, requiring only an additional parameter quantity of $D \times K$ and a relatively small amount of computational resources. Therefore, our proposed method offers promising prospects for large-scale image retrieval tasks.

## Ethics Statement

This paper primarily focuses on hash technology, aiming to enhance data compression and retrieval while extending current deep hash methods to large-scale scenarios. The datasets we utilize are publicly available, and we ensure strict adherence to their respective open licenses.

## Acknowledgement

This work is supported in part by the National Key Research and Development Program of China under Grant No. 2021ZD40303; in part by the Natural Science Foundation of China under Grant No. U20B2052, 61936011; in part by the National Key Research and Development Program of China under Grant No. 2018YFE0118400; in part by the National Natural Science Foundation Innovation Research Group Project under Grant No. 62321003.

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
