# Supplementary Material for Unleashing the Full Potential of Product Quantization for Large-Scale Image Retrieval

**Yu Liang**[1,4*]    **Shiliang Zhang**[2]    **Kenli Li**[1†]    **Xiaoyu Wang**[3†]

[1]College of Computer Science and Electronic Engineering, Hunan University
[2]National Key Laboratory for Multimedia Information Processing,
School of Computer Science, Peking University
[3]The Hong Kong University of Science and Technology(Guangzhou)
[4]Intellifusion Inc.
{leungyu, lkl}@hnu.edu.cn, slzhang.jdl@pku.edu.cn, fanghuaxue@gmail.com

## 1 Summary

This supplementary material provides further elaboration and discussion on our work, including additional details that support our findings.

## 2 More Detail on Trends of PQ Search

We provide the detailed of the PQ retrieval performance trends of Glint360K trained using the CosFace loss in Tab. 1. The result demonstrate that PQ performance rapidly degrades for short encoding lengths across various backbones.

Table 1: Comparison of the asymmetric retrieval performance trends of PQ with different backbone. 'L2' represents using the original features for L2 retrieval, while 'PQ256' represents dividing each feature into 256 segments for PQ retrieval, and 8 bits per segment.

| Glint360K | | | | | | | | | |
|---|---|---|---|---|---|---|---|---|---|
| Backbone | Metric | L2 | PQ256 | PQ128 | PQ64 | PQ32 | PQ16 | PQ8 | PQ4 |
| iResnet18 | Top1 | 0.9608 | 0.9604 | 0.9573 | 0.9500 | 0.9189 | 0.8005 | 0.4588 | 0.0973 |
| | Top5 | 0.9719 | 0.9717 | 0.9705 | 0.9657 | 0.9463 | 0.8659 | 0.5958 | 0.1940 |
| | Top20 | 0.9762 | 0.9762 | 0.9754 | 0.9723 | 0.9592 | 0.9025 | 0.6903 | 0.3031 |
| iResnet50 | Top1 | 0.9779 | 0.9778 | 0.9769 | 0.9754 | 0.9657 | 0.9184 | 0.6665 | 0.1724 |
| | Top5 | 0.9812 | 0.9812 | 0.9807 | 0.9800 | 0.9758 | 0.9492 | 0.7749 | 0.2999 |
| | Top20 | 0.9824 | 0.9824 | 0.9820 | 0.9814 | 0.9786 | 0.9610 | 0.8404 | 0.4254 |
| iResnet100 | Top1 | 0.9796 | 0.9795 | 0.9793 | 0.9783 | 0.9732 | 0.9465 | 0.7380 | 0.2074 |
| | Top5 | 0.9823 | 0.9823 | 0.9822 | 0.9815 | 0.9795 | 0.9658 | 0.8378 | 0.3569 |
| | Top20 | 0.9832 | 0.9832 | 9.9831 | 0.9824 | 0.9810 | 0.9720 | 0.8878 | 0.4941 |

---

[*]Work done as an intern at Intellifusion.
[†]Corresponding author.

37th Conference on Neural Information Processing Systems (NeurIPS 2023).

# 3 More Comparisons with Different Backbone Settings

To further reinforce the results of Table 2 in the main text, we present supplementary results for two highly competitive methods, OrthoHash[1] and GreedyHash[2], which have exhibited commendable performance on iResnet50. Specifically, we evaluated their performance on iResnet18 and iResnet100. The performance comparison is shown in the following Tab. 2. As can be observed, all methods achieve performance improvements as the network capabilities increase. And in comparison to these two excellent methods, our approach also achieves significant performance improvements.

Table 2: Under network structure iResnet18 and iResnet100 with various bits settings, our method achieved consistent and significant improvements.

| Glint360K iResnet18 | | | | | | | | | |
|---|---|---|---|---|---|---|---|---|---|
| Method | 32 bits | | | 64 bits | | | 128 bits | | |
| | Top-1 | Top-5 | Top-20 | Top-1 | Top-5 | Top-20 | Top-1 | Top-5 | Top-20 |
| PQ | 0.0973 | 0.1940 | 0.3031 | 0.4588 | 0.5958 | 0.6903 | 0.8005 | 0.8659 | 0.9025 |
| OrthoHash[1] | 0.0384 | 0.0528 | 0.0650 | 0.0960 | 0.1206 | 0.1421 | 0.2024 | 0.2426 | 0.2749 |
| GreedyHash[2] | 0.0991 | 0.1921 | 0.3014 | 0.4003 | 0.5507 | 0.6620 | 0.7344 | 0.8290 | 0.8824 |
| FPPQ(Ours) | **0.6945** | **0.7576** | **0.8006** | **0.7019** | **0.7890** | **0.8414** | **0.8231** | **0.8824** | **0.9133** |
| Glint360K iResnet100 | | | | | | | | | |
| PQ | 0.2074 | 0.3569 | 0.4941 | 0.7380 | 0.8378 | 0.8878 | 0.9465 | 0.9658 | 0.9720 |
| OrthoHash[1] | 0.4904 | 0.5511 | 0.5953 | 0.7133 | 0.7564 | 0.7865 | 0.7804 | 0.8185 | 0.8444 |
| GreedyHash[2] | 0.5902 | 0.7271 | 0.8035 | 0.4003 | 0.5507 | 0.6620 | 0.7344 | 0.8290 | 0.8824 |
| FPPQ(Ours) | **0.9305** | **0.9700** | **0.9734** | **0.9578** | **0.9728** | **0.9753** | **0.9679** | **0.9746** | **0.9772** |

# 4 Impact of Feature Preprocessing

We conducted a numerical evaluation of the performance of FPPQ on the Glint360K dataset, using different feature preprocessing methods. These methods include the direct use of original features, feature normalization, and feature segment normalization. The results presented in Table 3, indicate that our approach has a minimal effect on the impact of data preprocessing operations.

Table 3: Comparison of the performance of different data preprocessing operations at retrieval phase, where '$F$' denotes direct use of the original features, '$Fnor$' represents feature normalization, and '$SegFnor$' represents feature segment normalization.

| Glint360k iResnet50 | | | | |
|---|---|---|---|---|
| Preprocessing | Metric | PQ16 | PQ8 | PQ4 |
| $F$ | Top-1 | 0.9568 | 0.9467 | 0.9343 |
| | Top-5 | 0.9690 | 0.9571 | 0.9601 |
| | Top-20 | 0.9731 | 0.9628 | 0.9652 |
| $Fnor$ | Top-1 | 0.9571 | 0.9467 | 0.9343 |
| | Top-5 | 0.9690 | 0.9571 | 0.9601 |
| | Top-20 | 0.9731 | 0.9628 | 0.9652 |
| $SegFnor$ | Top-1 | 0.9563 | 0.9469 | 0.9332 |
| | Top-5 | 0.9687 | 0.9571 | 0.9604 |
| | Top-20 | 0.9730 | 0.9628 | 0.9655 |

# 5 Two-Stage Retrieval and Re-ranking

In certain scenarios, secondary retrieval may be required to further improve retrieval performance on the results of a previous search. For example, one can use PQ4 retrieval to obtain N returned samples and then perform L2 retrieval on these results to fine-tune the search. In this case, the L2 retrieval

performance becomes crucial. By incorporating a classification loss for the full features, our method considers this scenario. We evaluated the performance of our method on L2 retrieval and present the results in Table 4.

Table 4: The performance changes for PQ and L2 retrieval after applying our method.

| Glint360k iResnet50 | | | | |
|---|---|---|---|---|
| Method | | Top-1 | Top-5 | Top-20 |
| L2 | | 0.9779 | 0.9812 | 0.9824 |
| PQ4 | | 0.1724 | 0.2999 | 0.4254 |
| FPPQ(Ours) | PQ4 | 0.9342(+0.7618) | 0.9601(+0.6602) | 0.9652(+0.5398) |
| | L2 | 0.9735(-0.0044) | 0.9787(-0.0025) | 0.9805(-0.0019) |
| PQ8 | | 0.6665 | 0.7749 | 0.8404 |
| FPPQ(Ours) | PQ8 | 0.9467(+0.3801) | 0.9571(+0.1822) | 0.9628(+0.1224) |
| | L2 | 0.9735(-0.0044) | 0.9789(-0.0023) | 0.9806(-0.0018) |
| PQ16 | | 0.9184 | 0.9492 | 0.9610 |
| FPPQ(Ours) | PQ16 | 0.9568(+0.0385) | 0.9690(+0.0198) | 0.9731(+0.0121) |
| | L2 | 0.9747(-0.0032) | 0.9788(-0.0024) | 0.9809(-0.0015) |

We can see that compared with the huge improvement in PQ retrieval, the impact of our method on L2 retrieval is slight. These results imply that our approach can have widespread applicability in real-world scenarios.

## 6 An Implementation

To provide a clear depiction of our framework, we present a concise description of the algorithm flow as depicted in Alg. 1.

---
**Algorithm 1** The concise implementation of our method
---

1: **procedure** TRAINING
2:     Generate average features of the classes $\{\mathcal{F}_{avg}\}$
3:     Generate class-specific PQ label $PQ(\{\mathcal{F}_{avg}\})$
4:     **repeat**
5:         Sample $x$ from $\mathcal{I}$ with class label $y$ and PQ label $PQ(y) = [k_1, \cdots, k_m, \cdots, k_M]$;
6:         Forward propagation
7:         Back-propagation updates the parameters
8:     **until** Total epoch exceeds;
9: **end procedure**

10: **procedure** RETRIEVAL
11:     Define the PQ retrieval system;
12:     Substitute place the codebook $\mathcal{C}$ of the PQ retrieval system with the weights $\mathcal{W}$ of the FC layer in the PQ branch, $\mathcal{C} \in \mathbb{R}^{M \times K \times (D/M)}$, where $\mathcal{C}_{mk} = w_{mk}$;
13:     Standard PQ retrieval operation;
14: **end procedure**

---