# OpenReview forum: "Unleashing the Full Potential of Product Quantization for Large-Scale Image Retrieval"
_NeurIPS.cc/2023/Conference — NeurIPS 2023 poster_

### Official Review · Reviewer_hk7b · 2023-07-05

**Soundness:** 2 fair
**Presentation:** 2 fair
**Contribution:** 3 good
**Rating:** 5
**Confidence:** 3

**Summary:**

This paper presents a deep product quantization (PQ) method for approximate nearest neighbor (ANN) search. The motivation is to reduce performance degradation when short codes are applied to large-scale datasets with a large number of categories. The proposed method learns discriminative PQ subvectors by CosFace loss with pseudo-ground truth labels obtained by applying PQ to the set of mean vectors of all the classes. Experiments on Glint360K and ImageNet datasets show that the proposed method yields highly competitive or better performance than several existing ANN methods.

**Strengths:**

Table 1 shows that when using 32-bit codes for the Glint360k dataset with a large number of categories, the performance of existing ANN methods is poor, and the proposed method has a significant advantage.

The proposed method achieves the same or better accuracy as the existing methods even for ImageNet with a small number of categories.

**Weaknesses:**

A. Due to several ambiguities, I could not understand some important details about the experiment and wondered if the evaluation was done in a fair manner. Specific examples are listed below.

A1. Reading Section 5.1 I assume that Glint360K is used only for training and Mageface and Facecrub are used for testing. However, I find an explanation in line 260, "We will assess the retrieval performance using top-1, top-5, and top-20 accuracies on the Glint360K datasets," which is contradictory.

A2. Related to the above point, line 221 says that "for each individual, we have M images. We will test each image by incorporating it into the gallery of distractors and employing each of the other M-1 images as a probe.” Is there any reason to use such a tricky protocol? If the gallery and probe sets are Megaface and Facecrub, respectively, then both should contain images of the same individuals. So I don't see why they need to add images of Facecrub to Megaface. It seems to me that there is a concern that this protocol could introduce a dataset bias to some extent into the evaluation.


A3. I assume the "both datasets" in line 230 refers to Celeb-500k and MS1M-Retinaface, which comprise Glint360K, but it is hard to understand because there is no specific reference to them.


A4. Reading line 239, it appears that the same learning conditions are applied to all the methods. However, the appropriate learning conditions are generally different for different methods and have a significant impact on their accuracy, so I was concerned that this comparison is really fair.


A5. Is the feature preprocessing also applied to the baselines to be compared? Even if the performance of the proposed method does not change with or without preprocessing, the performance of the baseline to be compared may not. For fairness, comparisons with normalized features should also be reported.


A6. Sections 5.2 and 5.2.1 each describe learning conditions, and some of them, such as the learning rate, appear to be contradictory. Further clarification would be needed.


B. Some additional evaluations should be conducted to clarify the effectiveness of the proposed method.

B1. Looking at the ImageNet results (Tables 3 and 4) and the Glint360K results (Tables 1 and 2), the trend of the performance gap is very different. What causes the performance gap should be clarified. Specifically, although the proposed method does not always provide significant improvement over the baselines on ImageNet, it shows significant advantages on Glint360K when shorter codes are used, suggesting that the number of classes has a significant impact on the performance gap. If this is true, it should be interesting to analyze the performance gap for various number of classes.


B2. Another simple baseline to be compared would be a straightforward combination of fine-tuned feature extraction backbone + supervised quantization, i.e., first extract a deep feature of an image by using feature extraction backbone fine-tuned with a training data (e.g., Glint360K), and then use supervised quantization (e.g., [Wang et al., Supervised Quantization for Similarity Search, ICCV2016]) to encode the feature into a short code. Comparison to such an approach may clarify the effectiveness of the proposed method further.


C. The proposed method is not very novel, since many deep product quantization methods have been proposed in the past (e.g., [23, 28, 45, 49, 50] or [Jang et al., Generalized Product Quantization Network for Semi-supervised Image Retrieval, CVPR2020]).


D. Some minor problems

D1. How is the <||F_m||, C_mk> in Equation (9) is computed?

D2. The number of categories is large enough compared to previous standards in ANN searches, but the sample size assumed in this paper does not seem very appealing given that public datasets with 1B-scale samples (such as GIST1B) have been used since the early years.

**Questions:**

I hope the authors will resolve all the ambiguities I listed in A1-A6.

**Limitations:**

I could not find any discussion of the limitations of the proposed method. It might be a good idea to discuss the performance if there exist distribution shifts among the training, gallery, and probe sets.

---

> ### Author Rebuttal · Authors · 2023-08-09
>
> **Q1. Retrieval performance.**
>
> We evaluate the unseen retrieval performance on the Glint360k dataset: Glint360k is exclusively used for training, while Mageface and Facecrub are employed for testing.
>
> **Q2. Evaluation protocol.**
>
> In fact, the protocol we described is widely recognized and implemented across various studies. For more specific details, we would recommend referring to  [Kemelmacher-Shlizerman, Ira, et al. "The megaface benchmark: 1 million faces for recognition at scale." CVPR 2016].
>
> **Q3. "Both datasets".**
>
> Thank you very much for pointing out the issues in line 203. Indeed, as you mentioned, "both datasets"  refers to the clean Celeb-500k and MS1M-Retinaface datasets. We will make the necessary corrections accordingly..
>
> **Q4. Same learning conditions.**
>
> In fact, for these methods, we utilized the default settings provided by the Cisip project, if available, or relied on the publicly accessible projects provided by the original authors. Considering that these methods may not provide specific learning rate strategies for the datasets we test, it would be unfeasible to perform grid search or other methods to find the optimal learning rate strategy for the numerous methods we compared. On the other hand, these methods mostly are based on fine-tuning, using the default settings, such as an initial learning rate of 0.0001 and decreased to 0.00001, may not have a disruptive impact on the conclusions drawn.
>
> **Q5. Feature preprocessing.**
>
> For other methods, we incorporate their specific feature preprocessing techniques as defined by the respective authors. This includes applying feature normalization, which often results in improved performance, if the methods recommend or require it.
>
>
> **Q6. Learning rate contradictory.**
>
> Section 5.2.1 provides further details compared to Section 5.2. On the Glint360k dataset, the setup is as follows: as stated in the papers of the deep hashing methods we compare against, they utilize pretrained models to initialize network parameters and perform fine-tuning. As a result, these methods set their initial learning rate to 0.0001. However, our approach conducts training from scratch, utilizing only the PQ label provided by the pre-trained model. Therefore, our initial learning rate is set to 0.1.
>
>
> **Q7. Performance comparison on various number of classes.**
>
> Thank you. This is indeed an interesting issue. To assess the performance variations across different numbers of classes, we conducted the following experiment: we selected training data from Glint360k, which included the first 3k, 5k, 10k, 72,046 (20% of the total class number), 180,116 (50%), and 360,232 (100%) classes. It is crucial to highlight that in previous setting, we conducted unseen retrieval on Glint360k using Megaface and Facecrub datasets. Recognizing that the reduced diversity of training data may impact the performance of unseen retrieval, we conducted tests on the first 20W images from the Glint360k dataset to evaluate the learning performance of various methods. From each class, we extracted one image to construct the query set, totaling 2862 images, while the remaining data served as the gallery set. We provided the performance results for Top1 and Top20 in the Tabel 1(provided in pdf).
> We can observe that as the number of classes increases, OrthoHash gradually weakens its ability. GreedyHash achieves it's optimal performance when the number of classes is small. Our method consistently achieves high performance.  For greater clarity, we more provide a trend visualization in Figure 3(provided in pdf).
>
>
>
> **Q8. Another baseline.**
>
> Thanks you for your suggestion. Similar to PQ, SQ is also a non-deep hashing method. It aims to learn a transformation matrix that maps features into a discriminative subspace and utilizes CQ ([50]) for retrieval. However, as the source code for SQ is not available, we cannot guarantee an accurate reproduction within a limited timeframe. We will try to include this baseline in the future.
>
>
> **Q9. The difference with other deep product quantization methods.**
>
> As one of the most popular methods for quantization retrieval, there are indeed have many techniques aiming to improve the retrieval performance of PQ. However, it is worth noting that our method stands out by extending the applicability of deep hashing methods to large-scale datasets, which sets it apart from other methods. Moreover, in terms of methodology, our approach exhibits significant differences compared to others. DPQ ([23]) achieves self-learning of PQ codes by classifying the soft quantized features and employing two regularization terms. GPQ[Jang et al., Generalized Product Quantization Network for Semi-supervised Image Retrieval, CVPR2020] is similar to DPQ as it classifies soft quantized features and utilizes a metric learning strategy within subvectors. Similarly, PQN ([45]) constructs soft quantized features and learns based on triplet loss. PQM ([28]) utilizing quantized features for classification and optimizing quantization error through iterative updates. In comparison, our approach focuses on subspaces. For OPQN ([49]), it's codebook is predefined and limited by the feature dimensionality. CQ ([50]) is a non-deep feature quantization method.
>
> **Q10. How is the <||F_m||, C_mk> in Equation (9) is computed.**
>
> F_m represents the normalized result of the m-th sub-feature. C_mk refers to the k-th item of the m-th sub-codebook, where ||C_mk|| = 1. <•> can either be the cosine distance or the Euclidean distance. As the C_mk has constant feature norm, for F_m, regardless of whether or not the normalization is applyed, it always finds the same closest C_mk, in other words, assigned to same C_mk, denoted as C_mk*.
>
> **Q11. About GIST1B**
>
> GIST1B is a fixed feature representation set that cannot be learned. For deep hashing methods, they require adaptive learning to the features. Therefore, deep hashing methods usually do not include benchmark comparisons on GIST1B.

---

> > ### Comment · Reviewer_hk7b · 2023-08-16
> >
> > I thank the authors for their responses. Several of my concerns have been resolved.
> >
> > Q1-Q7. Now these points are clear to me. I would expect the authors to clearly explain and emphasize these points in their paper.
> >
> > Q8. The proposed method (like other existing deep feature quantization methods) can be viewed as a joint optimization of features and codes. Comparisons with separate optimization approaches would be desirable.
> >
> > Q9. I am still not fully convinced. As the authors mention, GPQ uses metric learning of subvectors, which is similar to the proposed method. The novelty (at least at a high level) is not yet clear to me. Due to this, the source of superiority of the proposed method also remains unclear.
> >
> > Q10. The notation here is somewhat confusing to me. $||x||$ reminds me of a norm, so $<||x||, y>$ looks like an inner product of a scalar $||x||$ and a vector $y$, which is confusing. If the author wants to mean normalized $x$ by $||x||$, I would recommend the authors consider other alternatives such as $\bar{x}$.
> >
> > Q11. I was not trying to say that GIST1B should be used for experiments. It just seems to me that it is not large enough to claim that it is large-scale.

---

> > > ### Author Response · Authors · 2023-08-20
> > > **Response to the comment of hk7b  [1/2]**
> > >
> > > We appreciate your comments, and we will make further efforts to address your concerns.
> > >
> > > **Q1-Q7.** Thank you, and we greatly appreciate your advice.  We will make sure to provide clear explanations in our paper.
> > >
> > > **Q8.**  Thank you for your suggestion. We have made efforts to reproduce SQ[1] as much as possible, and the relevant code will also be released along with our method. In fact, SQ involves a W-step operation during its learning process, which requires performing matrix multiplication with a large-scale matrix (number of samples * number of classes). This makes SQ impractical for datasets of the scale of Glint360k. Therefore, we conducted experiments on the largest feasible dataset, ImageNet1K, using the 2048-dimensional features generated by a pre-trained ResNet50 (as per our original paper's setup). The experimental results are as follows, clearly demonstrating the advantages of our method over SQ.
> > >
> > > |        | ImageNet1K | mAP@1000 |
> > > |:-----------:|:--------------:|:---------------:|
> > > | Methods |   32-bit   |  64-bit    |
> > > |SQ     |  0.3004    |  0.3599   |
> > > |Ours    |   **0.6207**   |  **0.6543**   |
> > >
> > > **Q9.** Indeed, as you mentioned, our method shares similarities with GPQ[2], such as we all learn a codebook. However, our method showcases distinct differences from GPQ in the following aspects:
> > > 1. GPQ utilizes softmax-based soft assignment of codewords to obtain quantized vectors, whereas our method does not involve this process.
> > > 2. GPQ does not incorporate a classifier for the complete feature; instead, it utilizes sub-vectors for category differentiation. In contrast, our approach includes a classifier for the complete feature, facilitating secondary retrieval against complete features.
> > > 3. The codebook in GPQ is mainly updated through soft assignment by a matrix $W_m$, while in our method, it is updated through direct loss backpropagation.
> > > 4. Our method learns a set of PQ labels to ensure the mapping of the same category to the same PQ code. In comparison, GPQ focuses on explicitly optimizing the similarity between quantized features and original features to reduce quantization error, using the N-pair Product Quantization loss.
> > >
> > > In practice, GPQ may not be well-suited for learning on large-scale data. The process of updating the Codebook through soft assignment based on the matrix $W_m \in \mathbb{R}^(M \times D \times Class\\_num)$ , where $M$ is the number of feature segments and $D$ is the feature dimension, becomes challenging as the $Class\\_num$ increases. In our experiments on Glint360k using the GPQ code provided by the authors, we encountered significant difficulties due to the extremely slow learning process of GPQ. Training alone would take approximately 492 hours, making it impractical to proceed with the experiments. To address this issue, we made slight modifications to the training process and introduced a variant called GPQ*. In GPQ*, we accelerated the training process by reducing the frequency of codebook soft assignment based on $W_m$, updating the codebook only every 100 steps instead of every step. We provide a runtime and performance comparison between our method and GPQ under the same experimental conditions (iResnet18, 8 * 3090Ti, 20 epochs, batch-size 128 * 8, SGD optimizer, 32-bit). The results are presented in the table below.
> > >
> > > |Method| Sample/sec  | Training Time  |  Top1 | Top20 |
> > > |:--------:|:--------------:|:-----------------:|:---------:| :-------:|
> > > | GPQ  |    ~192   |  ~ 492 hour  | -      |    -              |
> > > |GPQ*  |   ~1948  |  ~ 48 hour   | 0.0618  | 0.2105 |
> > > | Ours  |  **~6144**     |  **~ 15 hour**   | **0.6945**  | **0.8231** |
> > >
> > > The experimental results clearly demonstrate the significant advantages of our method over GPQ in terms of both learning efficiency and performance. Additionally, it can be observed that GPQ* seems to have poor performance. One possible reason for this is that the soft assignment of the codebook based on $W_m$  plays a crucial role in GPQ. Another reason could be that GPQ, to embed semantic information in codewords, utilizes sub-features for comprehensive classification (e.g., 360,232 categories), resulting in GPQ utilizing only a fraction of the feature length, namely $D/M$. It is well-known that the dimensionality of features directly affects their expressiveness. While this setup may not significantly impact the performance on small-scale datasets, as the data scale and number of categories increase, learning becomes more challenging, and the limitations of GPQ become apparent.
> > >
> > > **Q10.** Thank you very much for the suggestion. We will incorporate the change and use "$\overline{x}$" to represent the normalized x, instead of $||x||$.

---

> > > > ### Author Response · Authors · 2023-08-20
> > > > **Response to the comment of hk7b [2/2]**
> > > >
> > > > **Q11.** Indeed, the scale of the 17 million samples we tested in our experiments may still not be large enough, considering the explosive growth of data. However, it is noteworthy that our method has already achieved significant improvement compared to recently claimed methods for large-scale image retrieval. For instance, DGDH [3] has a maximum data size of 1,034,912 and a maximum number of classes of 91, OrthoHash [4] utilizes the ImageNet1K dataset, and OPQN [5] has a maximum data size of 834,300 and a maximum number of classes of 2,781, among others. In terms of data and class scale, our method has achieved an improvement of one to two orders of magnitude, respectively. Moreover, as mentioned in our response to Q8 and Q9, current hash methods often face limitations in scalability when applied to large datasets. However, our method overcomes this limitation as indicated in our response to **Q1** of review **kMZn**, where we state that the parameter and computational complexity of our method are not restricted by the scale of the dataset. This suggests that our method has the potential to be effectively applied to even larger-scale datasets. In our future work, we will further investigate the performance of our method on larger scales.
> > > >
> > > >
> > > >
> > > > **Reference**
> > > >
> > > > [1] Wang, Xiaojuan, et al. "Supervised quantization for similarity search." Proceedings of the IEEE Conference on Computer Vision and Pattern Recognition. 2016.
> > > >
> > > > [2] Jang, Young Kyun, and Nam Ik Cho. "Generalized product quantization network for semi-supervised image retrieval." Proceedings of the IEEE/CVF Conference on Computer Vision and Pattern Recognition. 2020.
> > > >
> > > > [3] Dong, Guohua, et al. "Discriminative geometric-structure-based deep hashing for large-scale image retrieval." IEEE Transactions on Cybernetics (2022).
> > > >
> > > > [4] Hoe, Jiun Tian, et al. "One loss for all: Deep hashing with a single cosine similarity based learning objective." Advances in Neural Information Processing Systems 34 (2021): 24286-24298.
> > > >
> > > > [5] Zhang, Ming, Xuefei Zhe, and Hong Yan. "Orthonormal product quantization network for scalable face image retrieval." Pattern Recognition 141 (2023): 109671.

---

> > > ### Author Response · Authors · 2023-08-21
> > > **Confirmation of our response**
> > >
> > > We would like to know if there are any further questions or suggestions. If our responses have addressed your concerns, would you consider raising the score? Once again, we appreciate your suggestions.

---

### Official Review · Reviewer_AU32 · 2023-07-05

**Soundness:** 3 good
**Presentation:** 2 fair
**Contribution:** 2 fair
**Rating:** 7
**Confidence:** 5

**Summary:**

In this paper, a framework for large-scale image retrieval is proposed, which is based on deep hashing and product quantization (PQ). The key contribution of this framework is the alleviation of the performance crash problem that arises when using very short PQ codes to save space and computation. The performance of the proposed method is demonstrated through experiments conducted on multiple large-scale datasets with numerous classes.

**Strengths:**

1.	This paper is well-organized and provides clear introductions to the details.
2.	It explains the potential reasons behind the performance decline of short PQ codes and supports its claims through extensive experiments.
3.	The proposed method is both novel and potentially practical, alleviating the problems of saving space and time in large-scale image retrieval.

**Weaknesses:**

1.	Several specific experiments, such as "different network structures," "visualization of sub-features and clustering centers," and "compression efficiency," are conducted using the proposed method and traditional PQ alone. However, this comparison may be unfair as it overlooks other outstanding methods.
2.	As the key contribution, it is important to provide a thorough comparison of the performance decline or crack between the proposed PQ method and other outstanding PQ-based methods when using shorter PQ codes.
3.	Visual retrieval samples of Top-K should be provided for clarity.

**Questions:**

Line 116 of Page 3: It is not clear whether "N"/"2^b" refers to "the class number of the dataset"/"256" based on the previous explanation (Line 113-115). Additionally, it should be noted that "N" is used to represent the batch size in Equation 2.

---

> ### Author Rebuttal · Authors · 2023-08-09
>
> **Q1: More results.**
>
> **Different network structures.** Thank you. Because of the limited time, we have supplemented the results of two outstanding methods, OrthoHash and GreedyHash, which have shown relatively good performance on iResnet50 in our paper. Specifically, we evaluated their performance on iResnet18 and iResnet100 as well (the experiment of OrthoHash with 64-bit on iResnet100 is still in progress, and we will provide the results later). The performance comparison is tabulated below. As can be observed, in comparison to these two excellent methods, our approach equally demonstrates substantial performance gains. Furthermore, all techniques exhibit performance improvements with an increase in network capabilities.
> .
> |     |     |      |     |iR18||||||
> |:---:|---:|:---:|:---|---:|:---:|:---|---:|:---:|:---|
> Method ||32 bits||| 64 bits |||128 bits
> ||Top-1| Top-5| Top-20 |Top-1 |Top-5| Top-20 |Top-1 |Top-5 |Top-20
> PQ  |     0.0973  |     0.1940  |     0.3031 |      0.4588 |      0.5958  |     0.6903  |     0.8005  |     0.8659  |     0.9025
> OrthoHash |      0.0384 |      0.0528 |      0.0650 |      0.0960  |     0.1206 |      0.1421 |      0.2024  |     0.2426  |     0.2749
> GreedyHash |      0.0991 |      0.1921 |      0.3014 |      0.4003 |      0.5507  |     0.6620 |      0.7344  |     0.8290 |     0.8824
> Ours |      **0.6945** |      **0.7576** |      **0.8006** |      **0.7019** |      **0.7890** |      **0.8414** |      **0.8231** |      **0.8824**  |     **0.9133**
>
>
>
> |     |     |      |     |iR100||||||
> |:---:|---:|:---:|:---|---:|:---:|:---|---:|:---:|:---|
> Method ||32 bits||| 64 bits |||128 bits
> ||Top-1| Top-5| Top-20 |Top-1 |Top-5| Top-20 |Top-1 |Top-5 |Top-20
> PQ | 0.2074 |0.3569 |0.4941 |0.7380| 0.8378 |0.8878 |0.9465 |0.9658 |0.9720
> OrthoHash | 0.4904 |0.5511 |0.5953 |- |- |- |0.7804| 0.8185 |0.8444
> GreedyHash | 0.5902       |0.7271 |0.8035 |0.7475	|0.8381	|0.8861	| 0.8529	| 0.9135	 |0.9415
> Ours |**0.9305** | **0.9700**       |**0.9734** |**0.9578** |**0.9728** |**0.9753** |**0.9679**| **0.9746** |**0.9772**
>
>
> **Visualization of sub-features and clustering centers.** Other binary deep hashing methods lack the incorporation of sub-features and sub-clustering centers. Additionally, executing PQ-based deep hashing methods on the Glint360k dataset presents challenges, including convergence issues in methods like DPQ([23]) and computational complexity concerns in OPQN([49]), ADSVQ([51]), and DCDH([48]),  and others. Consequently, we solely visualizing and comparing the results between PQ and our proposed method.
>
> **Compression efficiency.** On the iResnet50 architecture, we provide a comprehensive visual comparison in Figure 2.
>
> **Q2. Comparison of the performance decline.**
>
> Thank you. The original paper's Table 1 illustrates the decline in performance of other deep hashing methods. Particularly, when using longer codes like 128 bits, these methods already face difficulties and exhibit lower retrieval performance in large-scale datasets. Moreover, they show a substantial decline in performance as the code length becomes shorter. For instance, OrthoHash's Top-1 accuracy decreases from 0.6626 at 128 bits to 0.5526 at 64 bits and further drops to 0.3098 at 32 bits. Similarly, GreedyHash's Top-1 accuracy decreases from 0.8259 at 128 bits to 0.6102 at 64 bits and further diminishes to 0.3688 at 32 bits.  More lucid visualization can be find in Figure 2(provided in pdf).
>
>
> **Q3. Top-K visualization.**
>
> Thank you for your suggestion. In Figure 1(provided in pdf), we present visualizations of the Top-20 retrieval results on the Megaface and Facecrub datasets. This encompasses comparisons between various ways: 1) our method, which includes L2 retrieval and PQ4 retrieval (without feature processing, normalization, or segment normalization), and 2) the original L2 retrieval and naive PQ4 retrieval (with or without feature normalization). It can be observed that our method generates similar top-20 results under different feature preprocessing settings, further validating the insensitivity of our method to feature preprocessing.  Additionally, due to the introduced noise from quantization, we can observe a low overlap rate of Top-20 results between the PQ4 retrieval result and the L2 retrieval result.

---

> > ### Comment · Reviewer_AU32 · 2023-08-20
> > **Thanks for the reply**
> >
> > Thanks the authors for their reply.
> > All of the issues have been modified, and I tend to change my original decision to Accept.

---

> > > ### Author Response · Authors · 2023-08-20
> > > **Thanks for the comment**
> > >
> > > Thank you, and we sincerely appreciate your valuable suggestions and positive feedback.

---

### Official Review · Reviewer_WmwN · 2023-07-07

**Soundness:** 3 good
**Presentation:** 3 good
**Contribution:** 3 good
**Rating:** 6
**Confidence:** 4

**Summary:**

This paper presents a novel deep hashing framework based on product quantization. It is different from conventional PQ in learning a set of predefined PQ codes of the classes via a softmax-based differentiable PQ branch. The proposed method is validated to be effective on large scale datasets, including Glint360k.

**Strengths:**

This paper tackles the hashing for large scale datasets with millions of categories and hundreds of millions of samples, which may extend the range that hashing methods can be applied for.

The codebook is learned via PQ branch rather than clustering with the predefined class-level PQ labels as supervision.

The proposed method can be used for retrieval as the traditional PQ but with a low-slope decay of retrieval performance with decreasing code lengths.

The proposed method is validated on large scale datasets to be effective.

**Weaknesses:**

While the overall idea and implementation is simple, there are some implementation or adaptation details not clearly clarified. This includes the PQ code duplication removal in Sec. 4.2. There are also some adaptations when applying the method on the ImageNet100 and ImageNet1K datasets, for example using instance-level features and use OrthoHash to generate PQ labels. It is better to clearly specify the scenarios for different settings to avoid misunderstanding.

While the method section describes both symmetric and asymmetric retrieval, only the asymmetric results are reported.

Line 277, 'atrong' -> 'strong'. Line 284, 'we' -> 'We'.

**Questions:**

In Fig. 2, the results for the proposed method are only shown for PQ 32 onwards. How about the results with PQ64, PQ128 and PQ256?

**Limitations:**

No. One limitation may be the setting of class-level PQ labels, which relies on a pre-trained model.

---

> ### Author Rebuttal · Authors · 2023-08-09
>
> **Q1: Further detail.**
>
> Thank you for your suggestion, now we provide further detail, and they will all be added to the supplementary material later.
>
> **PQ code duplication removal.** We provide further explanations here.  For the sake of clarity, if we need to modify $n$ items of PQ codes to achieve deduplication, we refer to it as "**n-replace**." Initially, we perform product quantization training on class average features to obtain PQ codes for each class, with each PQ code consisting of $M$ segments. In addressing a recurring PQ code $P$,  we begin first attempt "1-replace." For every sub-vector $F_m$ associated with $P_m$, we compute its distance to the $K$ cluster centers of the m-th sub-codebook. This computation yields a total of $M \times K$ distances. We then proceed sequentially, seeking to substitute the PQ code with the index of the nearest cluster center, thereby achieving deduplication. If none of the $M \times K$ possibilities allow for substitution, we progress to the "2-replace".  First, we forcibly apply the "1-replace" to substitute the PQ segment with the index $k^*$ of the nearest m-th sub-codebook, and we maintain this alteration. Following this, we iteratively apply the "1-replace"  based on this condition. This iterative procedure is reiterated until a non-recurring PQ code is detected. In simple terms, we aim to find a replaceable PQ code with minimal item changes for repeated PQ codes while minimizing the quantization error as much as possible.
>
> **Instance-level features to generate PQ labels.** Here, "instance" represents the sigle sample's features, not the class average features. In this setting, to avoid having insufficient samples for PQ cluster training, we utilize the instance-level features to train the PQ algorithm and obtain the codebook of PQ. This codebook will be used for encoding the class average features.
>
> **PQ label generation in OrthoHash's approach.** OrthoHash hope to generate hash codes with a maximally expansive Hamming distance interval. This is achieved through a process of repetitive Bernoulli sampling. If a sampled hash code is found to be in proximity to the Hamming distance of a previously collected sample, it is discarded and the sampling is repeated. Subsequent to the generation of hash codes using the OrthoHash's Approach, we divide it into $M$ segments and transform them into PQ codes.
>
>
> **Q2: Asymmetric results.**
>
> We conducted experiments on symmetric retrieval, and as shown in the table below, our method still achieved excellent performance. We observed comparable retrieval performance among PQ4, PQ8, and PQ16, and the increase in code length does not have a positive impact on symmetric retrieval performance. We think this can be attributed to the heightened noise in symmetric retrieval resulting from the expansion of the number of PQ code segments.
>
> |||Glint360k iResnet100||
> :----:|:----:|:----:|:----:
> ||PQ4 |PQ8| PQ16
> Top-1 | 0.9387 | 0.9343 | 0.9341
> Top-5 | 0.9625 | 0.9609 | 0.9492
> Top-20 | 0.9647 | 0.9649 | 0.9565
>
>
> We also conducted experiments using different backbones, and the results are as follows. It can be observed that, similar to asymmetric retrieval, as the complexity of the model increases, the performance improves.
>
>
> ||iResnet100 |iResnet50 | iResnet18|
> :----:|:----:|:----:|:----:
> ||PQ4 |PQ4| PQ4
> Top-1 | 0.9387 | 0.9263 | 0.6175
> Top-5 | 0.9625 | 0.9441 | 0.6524
> Top-20 | 0.9647 | 0.9489 | 0.6868
>
> **Q3: Minor sentence error.**
>
> Thanks for pointing it out. We will correct it and review the paper carefully.

---

> > ### Comment · Reviewer_WmwN · 2023-08-19
> > **Thanks for the response**
> >
> > I thank the authors for the detailed response.
> >
> > The response addresses part of my concerns, especially the implementation detail. However, the question I asked is not addressed. The results for PQ 64/128/256 are still unclear to me. Judging from the Fig.2 in the rebuttal, the proposed method is likely to perform similar to PQ or even worse for PQ 64/128/256. This makes the contribution limited.

---

> > > ### Author Response · Authors · 2023-08-20
> > > **Thanks for the comment**
> > >
> > > Thank you for the reminder, and we apologize for inadvertently overlooking your question while we were transferring the content from our Word document to the webpage of OpenReview.
> > >
> > > Given that our integration of the PQ branch somewhat restricts the discriminative learning of features, our method may not exhibit improved performance compared to traditional PQ for PQ64, PQ128, and PQ256. However, it is important to note that Fig. 2 in the rebuttal demonstrates that the performance of traditional PQ64 already closely resembles that of L2 brute-force retrieval. This can be attributed to the already very small quantization error of traditional PQ, allowing only limited potential for improvement. For instance, when employing PQ64, PQ128, or PQ256, the 512-dimensional feature is divided into 64/128/256 segments, respectively, with the use of 256 cluster centers to partition the 8/4/2-dimensional subspace (512/(64/128/256) = 8/4/2). Accordingly, we consider the traditional PQ to be sufficient when the subspace dimension is small.

---

> > > > ### Comment · Reviewer_WmwN · 2023-08-21
> > > >
> > > > Thanks for the clarification.
> > > >
> > > > As I understand, the proposed method is better than traditional PQ under certain conditions. I would appreciate if the authors could include the details and criteria for such conditions in the paper. For example, does the condition change with respect to the dataset or the dimension of features?

---

> > > > > ### Author Response · Authors · 2023-08-21
> > > > > **Thanks for the comment**
> > > > >
> > > > > Thanks for your suggestions.
> > > > >
> > > > > Respect to the dataset: Compared to traditional PQ, we believe that our method does not have specific requirements in this aspect, as our learning objective is to find an appropriate feature representation for the data, which also aligns with the considerations in the Nearest Neighbors Search (ANNs) task.
> > > > >
> > > > > Respect to the dimension of features: Actually, we think that the application scope of our method is related to the intensity of feature compression. It is more suitable for scenarios with higher compression intensity and can effectively delay the performance degradation trend of traditional PQ. In general, higher compression strength in traditional PQ results in a larger quantization error, leading to a more noticeable performance gap compared to L2 brute-force retrieval. For example, if we apply traditional PQ64 to 4096-dimensional features, 256 centroids will quantize a relatively high-dimensional subspace (4096/64=64). In such cases, traditional PQ64 may show a significant decline in performance compared to L2 brute-force retrieval, and our method can unleash the potential of PQ under a relatively high compression strengths.
> > > > >
> > > > > Once again, we appreciate your suggestions, and we will incorporate the relevant discussions into the revised version of our paper.

---

### Official Review · Reviewer_kMZn · 2023-07-08

**Soundness:** 3 good
**Presentation:** 3 good
**Contribution:** 3 good
**Rating:** 6
**Confidence:** 3

**Summary:**

This paper discusses a new deep hashing algorithm based on product quantization, which effectively addresses the issues of high computational cost and low accuracy. The algorithm successfully learns predefined PQ codes for different classes, achieving concise, efficient, and distinguishable codes. It has been validated on multiple large-scale datasets, including ImageNet100, ImageNet1K, and Glint360k, and has shown significant improvements.

**Strengths:**

It addresses the issues of applying deep hashing to large-scale data by providing an easy-to-implement method that does not involve large-scale matrix operations.

**Weaknesses:**

Training with the combination of two branches may increase the complexity and training time of the model. Additionally, the performance of the method depends on the selection of predefined PQ class labels, which is not extensively discussed in the paper.

**Questions:**

Does the method rely on a specific data distribution or dataset size? Is it applicable to different types of datasets or larger-scale datasets?

**Limitations:**

It is not clear whether this framework is applicable to non-visual tasks or non-image datasets as the validation is conducted on image datasets. Furthermore, the performance of this method on larger-scale datasets, such as those exceeding the size of the existing validation sets, is not explicitly explained.

---

> ### Author Rebuttal · Authors · 2023-08-09
>
>
> **Q1:The increase of complexity and training time.**
> In fact, the number of parameters in the PQ branch is very small. Let $M$ denote the number of segments in PQ, $D$ denote the dimension of the embedded features, and $K$ denote the number of cluster centers. The number of parameters in the PQ branch is given by $M × (D/M) × K = D × K$, and the FLOPs (Floating Point Operations) is calculated as $M × (D/M + 1) × K = (D + M) × K$, which includes $D × K$ multiplications and $M × K$ additions. We have listed the specific computations and number of parameters in the following table. We assumed a relatively long code that we need to learn, where $M$ is set to 32, corresponding to learning 32 × 8 = 256-bit PQ codes, and $D$ is set to 512 with 256 cluster centers per PQ segment. As can be seen from the table below, the number of parameters and computations (FLOPs) of the PQ branch is very small compared to the backbone. It hardly increases the complexity and training time of the model.
>
> |  |  FLOPs  |       | Parameters |           |
> |--------|-------------|-----------|---|------|
> | |Backbone |PQ-Branch | Backbone  |PQ-Branch   |
> | iResnet18 |2634.12M |0.14M |24.03M |0.13M   |
> | iResnet50 |6346.83M |0.14M |43.59M |0.13M   |
> | iResnet100 |12149.32M |0.14M |65.16M |0.13M |
>
>
> **Q2: The select of the PQ label.**
> Our method seem is not particularly sensitive to the selection of PQ labels. We provide three pieces of evidence to support this claim:
>
> 1)In our experiments on ImageNet1K, as ImageNet1K has just 1000 classes, PQ could not be effectively trained due to the small number of categories. Moreover, the low discriminability between clAverage featureasses in the features generated by a pretrained model leads to a significant amount of duplicate PQ codes after PQ training and encoding. Therefore, instead of training on class average feature to generate PQ label, we utilize the OrthoHash's method to generate hash codes, whereby the hash codes are assigned randomly. These hash codes are then segmented and converted into decimal PQ code labels. The experimental results in Table 4 of the original paper demonstrate that our method also achieves good retrieval results.
>
> 2)Another experiment. We conducted involved using PQ4 labels generated by iResnet18 and iResnet100 to guide the training of iResnet50. The results of this experiment are presented in the table below. It can be observed that the retrieval performance in these settings is relatively consistent with the original setting. Note that using PQ codes generated by iResnet18 as labels resulted in better Top-1 performance compared to using PQ labels generated by iResnet100 and iResnet50. This observation aligns with the findings in Table 2 of the original paper, suggesting that PQ4 may have reached a performance bottleneck on the Glint360k dataset.
>
> |||Glint360k iResnet50||
> |:-------------------:|:---------:|:----------:|:-----------:|
> | Label  |    iR18-PQ4 | iR50-PQ4 | iR100-PQ4 |
> Top-1   | 0.9402 | 0.9342 | 0.9346
> Top-5   | 0.9611 | 0.9601 | 0.9609
> Top-20  | 0.9658 | 0.9652 | 0.9668
>
> 3)We also conducted an additional experiment to further discuss the choice of PQ labels. Actually, it is  a part of our future work. In this experiment, we aimed to move away from predefined PQ codes and instead utilized the weights of the model's fully connected layer as class prototypes to iteratively construct PQ codes during the training process. The obtained experimental results, shown below, demonstrate similar performance compared to the current methods.
>
> |||Glint360k iResnet50||
> |:-------------------:|:---------:|:----------:|:-----------:|
> |  |   PQ4 |PQ8 | PQ16 |
> Top-1   | 0.9369 | 0.9460 | 0.9682
> Top-5   | 0.9628 | 0.9702 | 0.9738
> Top-20  | 0.9676| 0.9723 | 0.9760
>
> **Q3: Rely on a specific data distribution or dataset size.**
> In terms of dataset size, our method has been validated on multiple large-scale datasets, including ImageNet100, ImageNet1K, and Glint360k. These datasets encompass a wide range of category sizes, ranging from 100 to 360k, and sample numbers ranging from 10k to 17 million, respectively. It is noteworthy that the Glint360k dataset utilized in our experiments is currently the largest dataset verified amongst all deep hashing methods, to the best of our knowledge. In terms of data distribution, the datasets used in our experiments consist of both face datasets and general object recognition dataset. Additional validation on even larger datasets, such as ImageNet21k and webface260k, will be left for our future work.

---

> > ### Author Response · Authors · 2023-08-21
> > **A answer to the limitations question**
> >
> > We appreciate your suggestions and we apologize for overlooking the limitations question while transferring the content from our Word document to the OpenReview webpage. We now to answer the question.
> >
> > **Q4: Whether this framework is applicable to non-visual tasks or non-image datasets.**
> > Thank you. Similar to some previous deep hash methods, this work also focuses on the visual domain. However, for the task of Approximate Nearest Neighbors Search (ANNs), we believe that the primary objective is to find a suitable representation for the data. In the future, we will validate our method on non-visual tasks as much as possible.

---

### Author Rebuttal · Authors · 2023-08-09

We thank all reviewers for their valuable comments. Reviewers kMZn, WmwN, and AU32 have mentioned that the method proposed in this paper is novel(new) and have affirmed its potential applicability. Reviewers hk7b also acknowledged the contributions of our method.

The PDF file contains Figure 1, Figure 2, Figure 3, and Table 1.

Moving forward, we will address the questions raised by the reviewers point by point.

---

### Decision · Program_Chairs · 2023-09-21

**Decision:**

Accept (poster)

**Comment:**

Overall, this paper presents a novel approach to address the challenges of applying deep hashing to large-scale datasets, specifically focusing on product quantization (PQ) as a key component. The proposed method offers several strengths, such as its ease of implementation, scalability to large-scale datasets with millions of categories and samples, and its potential practicality for space and time-efficient large-scale image retrieval. All the reviewers unanimously agree that the paper presents a strong contribution in the field of deep hashing over large-scale datasets. To strengthen the paper for publication, the authors should provide greater clarity on implementation details, and expand their comparison with the existing methods using shorter PQ codes. Additionally, including visual retrieval samples would further enhance the paper's comprehensibility. In summary, this paper is well-received by all the reviewers and can be accepted with minor revisions to address the mentioned concerns.